# Information Security Behavior in Health Information Systems: A Review of Research Trends and Antecedent Factors

**DOI:** 10.3390/healthcare10122531

**Published:** 2022-12-14

**Authors:** Puspita Kencana Sari, Putu Wuri Handayani, Achmad Nizar Hidayanto, Setiadi Yazid, Rizal Fathoni Aji

**Affiliations:** 1Faculty of Computer Science, Universitas Indonesia, Depok 16424, Indonesia; 2Faculty of Economic & Business, Telkom University, Bandung 40257, Indonesia

**Keywords:** information security behavior, antecedent factor, healthcare, health information system, systematic review

## Abstract

This study aims to review the literature on antecedent factors of information security related to the protection of health information systems (HISs) in the healthcare organization. We classify those factors into organizational and individual aspects. We followed the Preferred Reporting Items for Systematic Reviews and Meta-Analyses (PRISMA) framework. Academic articles were sourced from five online databases (Scopus, PubMed, IEEE, ScienceDirect, and SAGE) using keywords related to information security, behavior, and healthcare facilities. The search yielded 35 studies, in which the three most frequent individual factors were self-efficacy, perceived severity, and attitudes, while the three most frequent organizational factors were management support, cues to action, and organizational culture. Individual factors for patients and medical students are still understudied, as are the organizational factors of academic healthcare facilities. More individual factors have been found to significantly influence security behavior. Previous studies have been dominated by the security compliance behavior of clinical and non-clinical hospital staff. These research gaps highlight the theoretical implications of this study. This study provides insight for managers of healthcare facilities and governments to consider individual factors in establishing information security policies and programs for improving security behavior.

## 1. Introduction

The implementation of health information systems (HISs) by healthcare providers has positive value in properly managing healthcare information but also has negative impacts, such as security and privacy risks. HISs are vulnerable to violations of information security and privacy. Openness and connectedness with many heterogeneous stakeholders in the health network also increase these risks [1]. The healthcare industry lags far behind other sectors in terms of digital literacy and information security, making them a primary target [2]. Serious data breach incidents in the healthcare industry have occurred in health insurance institutions in the United States [3,4], health research institutes in the United Kingdom [5], providers of general laboratory testing services and specialized diagnostics in Canada [6], and hospital networks [7] and blood donor agencies in Singapore [8]. Security breaches target different types of healthcare organizations, although HIPAA Journal [9] states that 75% of data breaches occur in healthcare providers. Therefore, healthcare providers must maintain the confidentiality, availability, and integrity of patient health information [10,11,12] as part of their healthcare service delivery.

Several aspects can make the medical environment especially challenging to manage in terms of security. Healthcare has a larger risk of insider threats than the banking and insurance industries, which both hold and manage highly sensitive information [13]. The medical setting is strongly influenced by ethical considerations for various professions [14], affecting their decisions and behavior. Communication and trust issues between medical personnel and patients [15,16] play a fundamental role in patient care. Network expansion of healthcare service providers promotes the policy of sharing data between related parties [17], which increases the susceptibility of patient information transferred via electronic forms, including data ownership issues [18], responsibility for ensuring confidentiality [19], and responsibility for data integrity [20]. Health facilities are open public organizations [14], causing difficulties in access control and physical security [21], even though they have higher vulnerability to information security risks [16]. Insider threats posed by people with legitimate access to information systems can come from temporary staff, such as medical students, residents, or interns, who have the same need for access to medical data as permanent employees [14,16]. Most healthcare organizations do not prioritize information security in their resource allocation [14], as healthcare services are their primary business. Employees have different values and norms for information security [22,23,24] because it is often seen as hampering productivity in healthcare, especially in emergencies; thus, the level of negligence in security controls is relatively high [14]. In healthcare, there is not the same degree of worry or caution as in certain other sectors, including the banking industry [25]. These conditions emphasize that security behavior is a significant factor influencing healthcare organizations’ security effectiveness [26].

Health information is considered to be the most confidential information among other types of personal information [14]. It has a high value on the black market and, thus, becomes the target of organized criminal networks [27]. Some possible impacts include threats to patients based on their medical condition, financial losses and loss of resources, death, serious injury, illegal sales of limited medical equipment and medicines, loss of organizational reputation, and failure to achieve the organization’s mission and goals [28,29]. The most extensive health data breaches have occurred internally, with most incidents being errors and incidents of misuse [30,31]. Previous studies [22,32,33] have revealed cases of security breaches caused by human factors. Therefore, information security management in healthcare organizations should encourage good security behavior among employees and other related parties.

Information security behavior is essential in order to ensure that information assets are well protected [34]. Information-security-related behavior is defined as employee behavior in using organizational information systems, including hardware, software, networks, etc., that have security implications [35] as a function of the information security components defined by information security policy [36,37]. A previous study by Guo [35] classified security behaviors into four categories: (1) Security assurance behavior refers to the employee’s deliberate behavior to protect the organization’s information system, where this action is beyond policymakers’ expectations. (2) Security-compliant behavior refers to intentional or unintentional behavior that does not violate an organization’s information security policy, as policymakers expect. (3) Security risk-taking behavior refers to intentional employee behavior that can carry security risks for the organization’s information system, even if the employee has no motive for causing damage. (4) Security-damaging behavior refers to intentional employee behavior that can damage the security of an organization’s information system.

Security assurance and security-compliant behavior are considered desirable security behavior (DSB) because they can promote the effectiveness of information security designed by an organization. Meanwhile, security risk-taking and security-damaging behavior are considered undesirable security behavior (USB) that employees must avoid. In the healthcare context, most studies on security behavior have focused on factors that affect DSB, such as compliance with the Health Insurance Portability and Accountability Act (HIPAA)’s security and privacy rules or information security policy. Other studies have also investigated factors influencing USB, such as the intention to disclose patient information. Management can optimize the factors that drive DSB and anticipate the factors that drive USB. Therefore, it is necessary to understand the antecedent factors of both DSB and USB in the healthcare context.

Several previous studies conducted systematic literature reviews related to information security in the health context, such as [38,39], which focused on technical aspects and information security control. In comparison, systematic literature reviews related to information security behavior and culture [40,41,42,43,44,45,46,47] have not focused on the healthcare context. We found two articles [48,49] presenting systematic literature reviews concerning information security behavior in health organizations. The study by Page [48] discussed organizational culture in general but did not focus on healthcare organizations. The review by Yeng et al. [49] investigated healthcare professionals’ individual factors that can influence their information security practices, including psychological, social, cultural, and demographic factors. However, organizational factors also significantly influence information security practices and behaviors [50,51]. Thus, the present study aims to fill the gap in previous systematic reviews [49] by exploring individual and organizational factors that influence information security behavior in healthcare organizations.

In the literature on this research topic, the terms “information security” and “cybersecurity” are frequently used synonymously. Cybersecurity is related to the data in cyberspace, in contrast to information security, which is the protection of all information [52]. In smaller healthcare facilities, it is possible that HISs’ implementation will not always be online. HIS security risks include medical staff members directly disclosing patient information to their families. Therefore, this study focuses on information security behavior. We investigated the research trends and antecedent factors of information security behavior in the healthcare context involving various types of HIS users in healthcare organizations, including clinical staff, non-clinical staff, and patients. Specifically, we asked the following research question: “What are the research trends and antecedent factors of information security behavior in health information systems from organizational and individual perspectives?”

To answer this research question, we adopted a systematic literature review methodology. To conduct and report our review, we used the Preferred Reporting Items for Systematic Reviews and Meta-Analysis (PRISMA) statement [53]. PRISMA emphasizes methods through which researchers may guarantee the transparent and thorough reporting of systematic reviews [54]. PRISMA 2020 updates the PRISMA 2009 statement, which includes 27-item checklists, a flow diagram, and an explanation [53]. The choice of a systematic review will provide us the opportunity to inquire into present trends in the emphasis placed on security behavior, security threats, and the variables that affect how users behave while protecting health information.

This study is expected to have theoretical and practical implications. First, this study provides a systematic overview for researchers of antecedent factors of information security behavior, specifically in healthcare organizations. Second, this study determines the organizational and individual elements mapped to USB and DSB from HIS users. These findings can provide insight to managers in healthcare organizations to help them design information security policies and programs to prevent information security breaches, especially for internal threats. Third, this study can provide lessons for regulators to develop information security regulations in the healthcare industry—especially for information security governance and culture.

## 2. Materials and Methods

This study adopted the PRISMA 2020 framework (Appendix A) [53]. PRISMA has been used in previous studies in the field of information systems primarily related to health services, such as user acceptance of hospital information systems [55], security and privacy in electronic health records [38,39], and information security culture in general [44]. This shows that information system studies can also use PRISMA in the context of health and information security.

### 2.1. Eligibility Criteria

We determined four inclusion criteria (IC) for this study, as follows: (IC1) original scientific articles, including research articles, conference papers, and systematic reviews; (IC2) full-text articles available and written in English; (IC3) the research examines factors that influence information security behavior; (IC4) the research investigates health information protection in healthcare organizations. For removing irrelevant studies, the following exclusion criteria (EC) were applied: (EC1) articles duplicated in another repository; (EC2) articles that report on information security behavior from multisector organizations—not specifically in the healthcare sector; (EC3) studies that evaluate information security behavior without uncovering any antecedent factors; (EC4) studies that explore HIS security in organizations other than healthcare organizations.

### 2.2. Search Strategy

The second step was determining the sources of information, keywords, and journal repositories. The keywords used reflected three categories: terms related to information security, behavior, and health organizations. The keywords used in searching the repositories were as follows: (“information security” OR “cybersecurity”) AND (“behavior” OR “awareness” OR “compliance” OR “practice”) AND (“hospital” OR “clinic” OR “health”). Five journal repositories were used as sources of information: ScienceDirect, PubMed, SAGE, IEEE, and Scopus. We applied a filter for publication type to retrieve only journal articles and conference papers. To explore all possible studies, there was no publication time limit. The search process was carried out in February 2022 and focused on five databases: ScienceDirect, Medline/PubMed, SAGE, IEEE Xplore, and Scopus. We exported all of the search results into BibTeX or RIS files. We imported those files into Mendeley as a reference tool to check for duplicates and conduct further analysis.

### 2.3. Data Items and Synthesis

The next step was to analyze some attributes of the articles collected—namely, the author names, publication year, source type, name of the journal or conference, country of study or author affiliation, research methods, sample unit (i.e., respondent), healthcare organization type, variables used in the research model, and foundational theory. The selected studies focus on factors that influence the information security behavior of HIS users who have access to patients’ health data in healthcare organizations. Articles discussing information security behavior in organizations in general but covering the health industry were excluded. After reducing the duplicate results from the repositories, we screened the reports by examining their titles and abstracts. Furthermore, the examination was carried out by searching for full-text articles of some candidates and assessing whether the articles met the inclusion criteria. If a paper met the criteria, it was added to the selected studies. The results of the selected studies are summarized in a table (Appendix A).

## 3. Results

### 3.1. Study Selection

The search results from the specified databases returned 5573 studies with the defined keywords. Duplicate records were removed, resulting in 4677 records being screened in the next step. The title and abstract screening resulted in the exclusion of 4496 records with no mention of information security behavior in healthcare. Consequently, 181 articles were sought for retrieval, but 28 reports did not meet IC2 (no access to full text and not written in English). Next, 153 full-text articles were assessed for eligibility; 83 papers did not meet IC3 (no focus on factors influencing information security behavior), and 35 papers did not meet IC4. Performing the final step of the review resulted in 35 studies. Figure 1 shows the complete steps of the PRISMA workflow carried out in this study.

### 3.2. Study Characteristics

Figure 2 shows trends in research on information security behavior in healthcare from 2008 to 2021. We identified the first study published in 2008. One selected study in 2022 was excluded due to a lack of data to represent the year (until February 2022). The study trend increased significantly in 2020 (seven studies), which might have been a response to the COVID-19 outbreak. Healthcare providers had to change how to provide services to patients by adopting various technological solutions, which increased their vulnerability to cyberattacks [56]. During the COVID-19 pandemic, the most common cyberattacks in the health sector were ransomware and phishing attacks caused by human factors and a lack of security awareness [56]. The number of studies has doubled since 2020, but only two of the studies reviewed [57,58] mention COVID-19 in their discussion. The number of studies decreased slightly in 2021 (five studies) but was still higher than in previous years. Figure 2 shows the summary of selected studies for further analysis. The detailed list of selected studies is available in the Appendix A.

Of the 35 studies included in this review, we analyzed the distribution according to the countries where the studies took their samples or were conducted. Table 1 shows that developed countries dominate the studies related to information security behavior in healthcare organizations. Most of the studies involved respondents or participants from the United States (11 studies), Taiwan (five studies), the Republic of Korea (four studies), Germany (four studies), Malaysia (two studies), Saudi Arabia (two studies), Norway (one study), and Spain (one study). One study took samples from Ireland, Italy, and Greece. There were only four studies from developing countries: South Africa (two studies), India (one study), and Indonesia (one study). The categories of developed and developing countries used in this study refer to their gross national income per capita per year as calculated by the World Bank Atlas [59].

Regarding the organization type, most studies were conducted in hospitals. Table 2 shows that 23 studies examined information security behavior in hospitals only. Five studies involved hospitals and other healthcare providers, such as private clinics, physical therapy facilities, mental healthcare facilities, nursing homes, public health centers, and physicians’ offices. Two investigated nursing schools, and two investigated academic medical centers. In the remaining three studies, the type of healthcare organization was not specified.

Table 3 shows the study characteristics according to the respondents or participants. Most of the studies involved clinical staff (25 studies), such as doctors, dentists, nurses, pharmacists, physical therapists, and nutritionists. Twenty-one studies involved non-clinical staff as respondents, such as administration staff, information technology (IT) staff, human resources experts, privacy officers, top-level management, and psychologists. In addition to the permanent staff of healthcare organizations, five studies investigated the information security behavior of temporary staff, such as medical students and interns. A single study took patients as respondents to measure their behavior in protecting personal information managed by medical facilities.

The research methods (Table 4) were primarily quantitative, surveying respondents through questionnaires (27 studies). Some studies complemented their surveys with experiments to observe actual behavior. Seven studies used qualitative methods—both empirical (i.e., interview) and analytical (i.e., literature review and conceptual models). Meanwhile, two other studies used mixed methods (i.e., survey and interview).

Table 5 shows where the selected studies were published. Most of the selected studies were journal articles (25 studies). Three sources contained more than one selected study. Meanwhile, nine studies were published in conference proceedings, with two of these sources containing more than one selected study.

Table 6 defines 20 distinct theories adopted as foundational in the selected studies. Most studies used a combination of two or more theories. The theories used in multiple studies were the theory of planned behavior (TPB; 10 studies), general deterrence theory (GDT; nine studies), protection motivation theory (PMT; eight studies), health belief model (HBM; five studies), and theory acceptance model (TAM; four studies). The TPB explains that social pressure and cognitive thinking influence individual behavior [86]. GDT describes how security behavior is influenced by deterrence beliefs and fears [87]. PMT is involved in the development of the HBM, which explains how individuals carry out a cognitive evaluation to determine appropriate behavior based on the ability to deal with threats [88,89]. The TAM provides a model of how people come to acknowledge and utilize technology [90]. However, the TPB was only adopted in studies related to DSB, while other frequent theories were adopted in both DSB and USB research.

Table 7 depicts the variance in the types of information security behavior examined in the selected studies. DSB was the most observed behavior (25 studies), with behavioral concerns with respect to compliance with information security policy and regulations (17 studies) or performing security protection according to best practices (eight studies). USB was examined in seven studies, with concerns including risky security practices (four studies) and information security policy violations (three studies). Meanwhile, three studies investigated security behavior with respect to both secure and insecure practices among HIS users.

### 3.3. Security Threat Model

A healthcare facility bases its information security policy on the security risk profile of the organization. The risk can be determined from security threats that may occur in the organization or refer to similar organizations as benchmarks. Previous studies [91] revealed that the most critical security threat in an HIS is a power failure, followed by human error and technological failures. Other studies [32,92] identified that most security threats were related to human behavior, such as password sharing, missing records, email misrouting, theft on the premises, procedures not followed, and the establishment of improper HIS privileges.

The selected studies also mention some threats and vulnerabilities to be addressed by improving information security protection by modifying the healthcare staff’s behavior. Since this systematic review focuses on the information security behavior of HIS users, most of the selected studies only show possible threats posed by insiders. We modeled the threat from selected studies by referring to [93] in breaking down the threat action, health information assets, vulnerabilities, and potential control actions. Threat action and control were classified based on ISO 27799:2016 [14] as the information security standard for health information. Figure 3 depicts various types of threats to health information, especially with insiders as the source. The number in the bar shows the number of selected studies mentioning the threat.

Here, we discuss the top three security threat actions discussed in the selected studies. The greatest security threat is the unauthorized use of the HIS (11 studies). This threat can lead to incident events because of vulnerabilities in the healthcare facilities—for example, lack of security awareness and policy compliance [11,50,58,70,81,82], use of multiple entry points to access electronic medical records [49,65] and forgetting to log out after using the HIS at an unattended workstation [85]. The second-greatest threat is masquerading by insiders, such as staff accessing the HIS without using their own account (seven studies). The vulnerabilities that can be exploited by this threat are weak information security policy compliance [57,81], weak access control management [67,83,84,85], and sharing of workstations to access the HIS [25]. The third-greatest threat is user error in handling information (six studies). This threat can be triggered by the weakness of information security policy compliance [57,74], ignorance of the risk involved [11], poor security skills and security monitoring [1], low user education, and lack of awareness of information security [50,75].

There are some actions that cannot be classified into threat types according to ISO 27799:2016 Annex A [14]. An example would be a nurse intentionally disclosing a patient’s health information to their family [64,77,79] with the assumption that this would make the medical treatment more efficient and benefit the healthcare facility. Meanwhile, an operation error in ISO 27799:2016 [14] refers to the unintentional disclosure of confidential information. Some selected studies [26,51,61,66,72,76] do not mention the threat action specifically but only describe a violation of the information security policy or regulation and health information leakage in a healthcare organization.

### 3.4. Antecedent Factors of Security Behavior

Antecedent factors were gathered from research variables that were proven to be significant in empirical studies included in this review. Of the 35 selected studies, four were conceptual studies and, thus, were excluded from the analysis. There were 59 different variables as antecedent factors that significantly influence information security behavior directly and indirectly. The number of variables shows enormous variation in information security behavior research in healthcare. The variables are also related to the various foundational theories in the selected studies. Some factors are derived from frequent foundational theories, i.e., the TPB, PMT, GDT, and HBM. This shows that information security behavior studies are likely to use approaches from psychology (TPB and PMT), criminology (GDT), and public health (HBM) [94].

Meanwhile, factors adopted from the information system domain (TAM) are mostly insignificant in influencing security behavior. These variables were grouped into individual and organizational factors and then mapped into two types of security behavior. Human factors in cybersecurity are better viewed from various perspectives. Some previous studies [51,61] agree that employee security behavior can be influenced by two types of factors—namely, organizational factors and individual factors.

#### 3.4.1. Individual Factors

Individual or personal factors investigate the individual reasoning and decision-making behind security behavior [95]. This study identified 31 distinct individual factors (Table 8) from the selected studies that empirically influence information security behavior. Fifteen factors appear in multiple studies. Four of them influence DSB and USB, examined in different studies.

The most frequent individual factor in the selected studies was self-efficacy (12 studies) derived from PMT. Almost half of the desirable security behavior studies observed that self-efficacy positively and significantly influences information security behavior directly [1,23,51,57,61,72,74,75] and indirectly [62,63,70], through other variables (e.g., perceived behavioral control and avoidance motivation). The other most frequent factors were perceived severity (10 studies) and perceived susceptibility (4 studies). Perceived severity positively influences security compliance behavior [65,71,74,75,81] and assurance behavior [62,63] or negatively influences damaging behavior [76]. Perceived susceptibility also positively influences compliance behavior [65,71,74] and assurance behavior [63,76]. Perceived susceptibility in some studies is called perceived vulnerability [71,76,78]. According to PMT and the HBM, these factors are components of threat appraisal, which explains people’s assessment of a security threat or risk that they will manage [96]. Some selected studies used the terms perceived threat [63] and perceived risk [65] to reflect healthcare staff’s perceptions of the security threat or risk according to their perceived severity and susceptibility, which then significantly influence their further security behavior intentions.

Perceived benefit (six studies) and perceived barriers (three studies) are also adopted from HBM constructs. A previous study [71] that adopted PMT used different terms to reflect perceived benefits and perceived barriers: response efficacy and response cost, respectively. Other words with similar meanings to perceived benefit and perceived barriers are safeguard effectiveness [63] and safeguard cost [63,65], respectively. Different studies [70,81] that adopted the TAM used the perceived usefulness construct but adopted a similar definition of perceived benefit in the context of security behavior.

The TPB, as the dominant foundational theory in the selected studies, also contributes to frequent factors—namely, attitudes (seven studies), subjective norms (seven studies), and perceived behavioral control (four studies). Attitude is commonly used as a mediating variable to predict health staff’s DSB based on individual and organizational factors. Perceived trust is frequently related to behavioral intentions in TPB studies [1,51,61,74].

Security awareness (seven studies) is adopted from the variable GDT [87] as a factor that deters people from engaging in undesirable behavior. Some studies used the general term information security awareness as a research variable [57,58,62,67], while others used health information security awareness, consisting of general and health-related issues, regulations, and relevant consequences [64,77].

Perceived responsibility (two studies) and personal norms (two studies) are individual factors that appeared more than once in studies related to DSB and USB. Perceived responsibility emphasizes that it is one’s job to achieve professional goals [79]. Meanwhile, personal norms define health staff’s values, such as perceiving an information security policy violation as inappropriate and unacceptable [58]. This value negatively influences the intention to disclose information [77] and positively influences attitudes toward information security policy compliance [58].

In examining HIS users who participated in the selected studies, we found that individual factors from patients have not yet been explored. One study that took patients as participants [69] only investigated organizational factors (i.e., data collection processes, secondary use, and system error) that can influence their security behavior. There are three factors that significantly influence information security behavior among both clinical and non-clinical staff of healthcare organizations and medical students: perceived severity, perceived susceptibility, and information security awareness. The other individual factors significantly influence one or two user types. Therefore, those factors can be explored in future research.

#### 3.4.2. Organizational Factors

Organizational factors investigate organizational issues—such as procedures, programs, work environment, and security culture—that can influence employees’ security behavior [50]. There were 26 distinct organizational factors (Table 9) that empirically affect information security behaviors in the selected studies. Six factors were identified in more than one study; three appeared in both USB and DSB studies. Fourteen factors were only examined in DSB studies, while seven were examined only in USB studies.

The most frequent organizational factor was management/organizational support (four studies). Previous studies [1,26,61,74] found that management support indirectly influences users’ behavior through various individual factors, such as perceived benefit, severity, self-efficacy, and trust. Management support can be measured through information security policy implementation, security training, and leadership from the top-level management [74].

Cues to action (three studies) are derived from the HBM construct. In selected studies [62,72,75], cues to action had a positive and significant influence on security behavior intention—mainly for security protection and compliance. None of the selected studies examined the effects of cues to action on the desire to commit a security violation or human error. A survey by Kessler et al. [66] measured organizational culture through practice, importance, and laxness, while Dong et al. [58] examined organizational culture in terms of top-level management beliefs and organizational control of information security issues.

The following factors appeared in two studies: Perceived certainty is derived from GDT, which can examine different acts or processes, such as detection [80] and punishment [73]. Two selected studies evaluated the impacts of peer influence and superior influence on different types of security behavior: protection intention [70] and non-compliance intention [82]. Both studies revealed that peer and superior influences significantly affect security behavior intentions through individual factors as mediating variables, such as subjective norms [70] and neutralization techniques [82].

Importantly, most of the selected studies took place in hospitals, and organizational factors mostly influence security behavior in a hospital context. Management support is the only factor that impacts all types of healthcare organizations. These results support the findings of previous studies [1,26,61,74], illustrating that support from management—such as information security policymaking—is the most important thing for all types of health organizations. However, in the selected studies, management support to deter undesirable security behavior was not investigated.

## 4. Discussion

Studies on information security behavior in healthcare organizations are still dominated by investigations into why people intend to comply with an organization’s information security policy or health security regulation, such as HIPAA. The most frequently adopted theory is the TPB, but the most frequent significant factors are derived from PMT as an improvement from the HBM. Attitudes, subjective norms, and perceived behavioral control as the constructs of the TPB were only investigated in DSB studies and were mostly combined with other theories, such as PMT and GDT. It is possible to explain human errors and violations by examining the staff’s attitudes toward information security behaviors [95]. However, the attitude was not a research variable in the selected studies related to USB.

The results empirically reveal that more individual (32 factors) than organizational (26 factors) aspects significantly affect information security behavior in the healthcare context. Those factors might positively (i.e., promoting) or negatively (i.e., preventing) affect the related behavior. This is consistent with the most frequently adopted foundational theories, the TPB and PMT, which focus on individual aspects of behavior. Although only two selected studies [50,82] explicitly segregated individual and organizational factors, many (16 studies) also examined both factors. Ten studies only used individual factors, while four studies only used organizational factors as significant antecedents to predict users’ security behavior. Hence, organizational aspects remain underexplored in this research field. However, most studies indicated that organizational factors significantly impact security behavior, mediated by individual factors.

Self-efficacy is the most significant individual factor that is only important in influencing DSB. A USB study [64] and a combined USB–DSB study [76] examined this factor. However, self-efficacy was not significant in predicting insecure behavior, such as the intention to disclose information and violate security controls. The other frequent individual factors were from PMT and the HBM: perceived severity, perceived susceptibility, perceived benefit, and perceived barriers. Perceived severity and perceived susceptibility can be influenced by the security awareness of healthcare staff [76], which reflects their knowledge and understanding of potential security issues and their consequences—both general and health-information-specific [77]. Together with perceived benefits and perceived barriers, self-efficacy compiles a construct known as coping appraisal, which affects information security intention [78]. Many studies measured the benefits of security protection using various terms, including perceived benefit, perceived usefulness, and response efficacy. Although they used different names for the variables in different contexts, they referred to the same definitions.

Management support, as the most significant organizational factor, is derived from GDT’s constructs. None of the selected studies examined management support as an antecedent factor of USB. Management support, such as providing security training to improve staff’s security awareness, can also influence self-efficacy [1,64,74,76]. Therefore, security managers in healthcare organizations can design some security policies and programs that facilitate the staff’s adoption of security practices and increase their confidence. Strengthening employee self-efficacy may increase the likelihood of effective security compliance. The next most significant organizational factor was cues to action from the HBM. The selected empirical studies showed that health staff’s security behavior could be predicted directly by cues to action, such as security campaigns and the influence of peers and superiors, which can promote security protections and compliance.

Some studies used demographic characteristics as differentiating factors, such as gender [66,72,80,81], age [25,66], occupation type [25,61,66], organization type [61,81], education [25], working duration [74,78,80]. However, these demographic differences were only found in DSB studies. Organizational and occupational characteristics can influence the self-efficacy of healthcare professionals in complying with privacy and security rules due to their different work environments [61]. Figure 4 depicts a summary of the antecedent factors of security behavior based on the selected studies.

The theoretical contributions of our research complement prior studies by adding and mapping previous inquiries to understand related factors, actors, providers, and behavior types. A systematic literature review by Yeng et al. [49] examined psychological, social, and cultural aspects of information security behavior. The study did not define individual and organizational factors as predictors of information security behavior. Moreover, the study only investigated general healthcare professionals’ perspectives as HIS users without including patients and other stakeholders among the healthcare providers. The COVID-19 pandemic has driven healthcare facilities to develop digital health approaches, such as telehealth, mobile health applications, and the Internet of Medical Things (IoMT). These initiatives can accelerate the exchange of health information by empowering patients to manage and share their medical information with various healthcare organizations. Patient-centered information exchange also requires the patient to play an active role in information security and privacy protection [97]. A previous study [69] investigating patient behavior did not examine individual factors.

The practical implications of our research provide lessons for decision-makers in healthcare organizations and governments to encourage the expected security behavior. The most frequent information security hazards in healthcare organizations are improper usage, insider impersonation, and human error when handling information. By considering specific elements such as self-efficacy, perceived severity, and information security knowledge, healthcare organizations may build security policies to reduce the occurrence and effects of these risks. For instance, educating users about the threats to information security and enhancing their technical skills to defend information security are only two examples of how to do this. For information security protection to be successful, it is also necessary to enhance organizational factors that can promote information security behavior, such as support and commitment from top-level management, peer and superior influence, and a positive corporate culture.

A limitation of this review is that we only analyzed the empirical studies to define significant antecedent factors and classify them as an individual or organizational factors. The most frequent factors were measured not by their appearance as research variables in the selected studies but by how many studies identified those factors as predictors of security behavior. Since the research methods of the empirical studies varied, this review could not determine the influence of each factor on the dependent variables. Therefore, the most frequent factors do not necessarily represent the most significant factors in evaluating health staff’s information security behavior. Previous studies revealed no established general model for information security behavior in healthcare. This study does not propose a specific model but, rather, shows the research gap for further investigation. Further research is necessary to learn more about the influencing factors among user groups in various healthcare organizations. Patients should be involved as research objects to determine how healthcare facilities should involve them in controlling information security.

## 5. Conclusions

Healthcare providers other than hospitals are understudied. Studies related to both DSB and USB show that the factors preventing protection can differ from those that promote information security violations. Therefore, future studies should investigate both types of security behavior. The development of technological solutions used by health facilities since the COVID-19 outbreak, such as telemedicine and mobile health applications, has expanded HIS coverage. Protecting health information security relies on healthcare professionals and patients participating in managing their data. Information security risks come not only from internal users at the healthcare provider but also from external users who have access rights to the system. Therefore, studies on information security behavior in healthcare organizations need to understand the patient’s perspective, which is still rarely studied.

## Figures and Tables

**Figure 1 healthcare-10-02531-f001:**
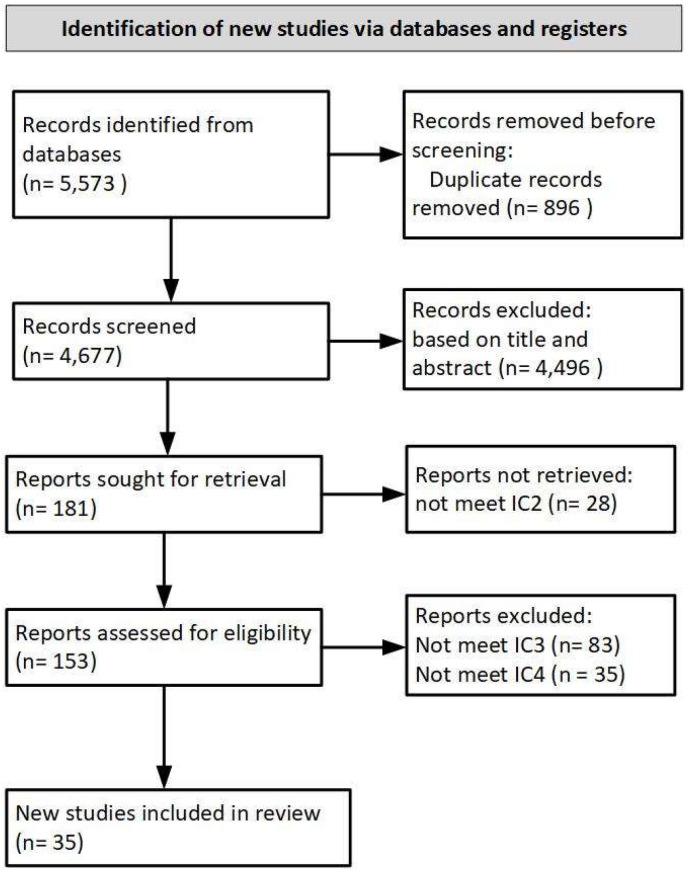
PRISMA workflow diagram (IC = Inclusion Criteria).

**Figure 2 healthcare-10-02531-f002:**
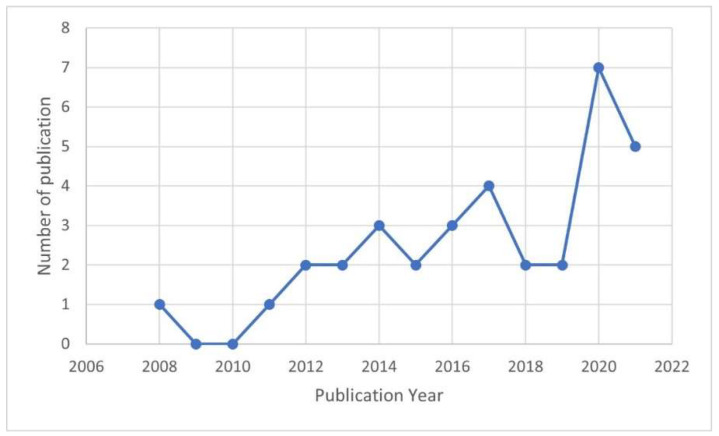
Research trends.

**Figure 3 healthcare-10-02531-f003:**
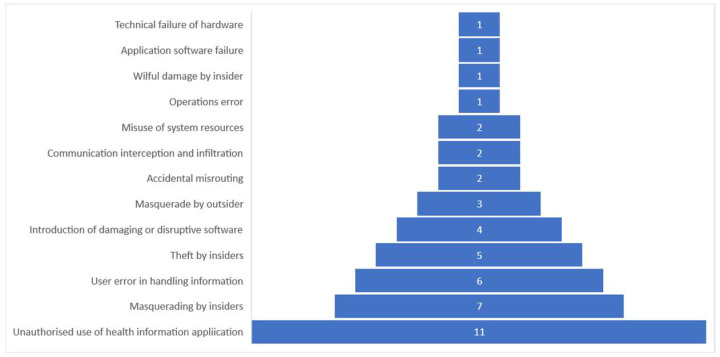
Threat actions were discussed in the selected studies.

**Figure 4 healthcare-10-02531-f004:**
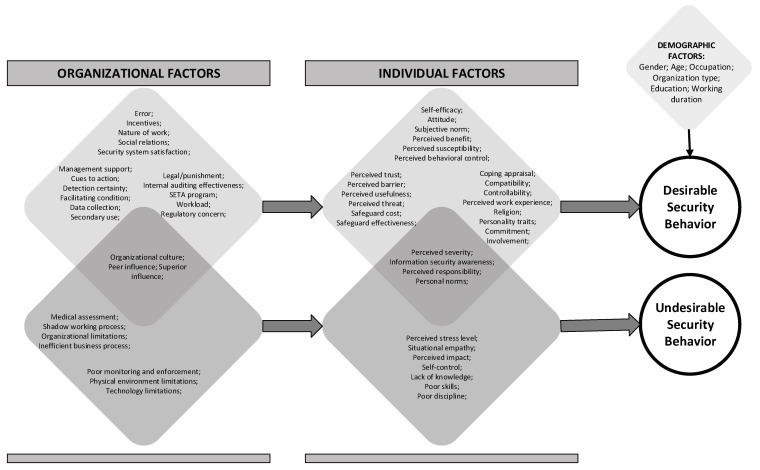
Antecedent factors of information security behavior in healthcare organizations.

**Table 1 healthcare-10-02531-t001:** Countries involved in the selected studies.

Category	Country	Frequency	Citation
Developed country	United States	11	[26,50,60,61,62,63,64,65,66,67,68]
	Taiwan	5	[69,70,71,72,73]
	Republic of Korea	4	[1,58,74,75]
	Germany	4	[76,77,78,79]
	Malaysia	2	[80,81]
	Saudi Arabia	2	[57,82]
	Norway	1	[49]
	Spain	1	[25]
	Ireland	1	[83]
	Italy	1	[83]
	Greece	1	[83]
Developing country	South Africa	2	[11,84]
	India	1	[51]
	Indonesia	1	[85]

**Table 2 healthcare-10-02531-t002:** Types of organizations involved in the selected studies.

Type of Organization	Frequency	Citation
Hospitals only	23	[1,25,49,51,57,58,60,63,67,68,69,70,71,72,74,75,76,78,80,81,82,84,85]
Hospitals and other providers (clinics, health centers, etc.)	5	[61,65,66,73,83]
Healthcare organizations (unspecified)	3	[11,50,79]
Nursing schools	2	[64,77]
Academic medical centers	2	[26,62]

**Table 3 healthcare-10-02531-t003:** Respondents involved in the selected studies.

Respondents	Frequency	Citation
Clinical staff (physicians, nurses, pharmacists, etc.)	25	[1,11,25,49,57,58,60,61,62,63,65,66,67,68,70,73,74,75,76,78,80,81,83,84,85]
Non-clinical staff (Administration staff, top-level management, IT staff, etc.)	21	[1,11,25,26,50,51,57,60,61,65,71,72,73,74,75,76,80,81,82,83,85]
Temporary staff (nursing students, interns)	5	[64,65,77,79,82]
Patients	1	[69]

**Table 4 healthcare-10-02531-t004:** Research methods of the selected studies.

Research Method	Frequency	Citation
Quantitative (survey, experiment)	26	[1,25,26,57,58,60,61,62,63,64,65,66,69,70,71,72,73,74,75,76,77,78,79,80,81,85]
Qualitative (interview)	3	[50,82,83]
Literature review	4	[11,67,84,85]
Mixed methods (interview and survey)	2	[51,68]

**Table 5 healthcare-10-02531-t005:** Source of the selected studies.

Source	Name of Publication	Frequency
Journal article	Health Information Management Journal	3
	Computers & Security	3
	INQUIRY: The Journal of Health Care Organization, Provision, and Financing	2
	Sustainability	1
	Symmetry	1
	International Journal of Environmental Research and Public Health	1
	International Journal of Medical Informatics	1
	Information Systems Research	1
	Information Management & Computer Security	1
	Health Informatics Journal	1
	BMC Medical Informatics and Decision Making	1
	International Journal of Health Care Quality Assurance	1
	Information Systems Frontiers	1
	Information Systems Journal	1
	Malaysian Journal of Computer Science	1
	European Journal of Information Systems	1
	JMIR Human Factors	1
	Journal of Medical Internet Research	1
	Security Journal	1
	Journal of Public Health	1
Proceedings	Procedia Technology	2
	Americas Conference on Information Systems (AMCIS)	2
	IEEE Conference on e-Learning, e-Management, and e-Services (IC3e)	1
	Hawaii International Conference on System Sciences	1
	International Conference on Information and Communication Systems (ICICS)	1
	International Conference on Availability, Reliability, and Security	1
	Conference on HCI for Cybersecurity, Privacy, and Trust	1
	Conference on Risks and Security of Internet and Systems	1

**Table 6 healthcare-10-02531-t006:** Foundational theories in the selected studies.

Foundational Theory	Frequency	Citation
Theory of planned behavior (TPB)	10	[1,49,51,57,60,61,62,70,74,80]
General deterrence theory (GDT)	9	[11,57,64,65,68,73,76,77,80]
Protection motivation theory (PMT)	8	[49,57,65,69,71,76,78,81]
Health belief model (HBM)	5	[49,62,72,74,75]
Theory acceptance model (TAM)	4	[61,65,70,81]
Social cognitive theory (SCT)	1	[84]
Norman’s action theory (NAT)	1	[50]
Concern for information privacy (CFIP)	1	[69]
Theory of reasoned action (TRA)	1	[71]
Power style theory (PST)	1	[51]
Social exchange theory (SET)	1	[51]
Technology threat avoidance theory (TTAT)	1	[63]
Unified theory of acceptance and use of technology (UTAUT)	1	[61]
Social control theory (SCoT)	1	[49]
Rational choice theory (RCT)	1	[57]
Social bond theory (SBT)	1	[58]
Cognitive moral development theory (CMDT)	1	[57]
Diffusion of innovation (DOI)	1	[57]
Prosocial rule breaking (PSRB)	1	[79]
Neutralization theory	1	[82]

**Table 7 healthcare-10-02531-t007:** Security behaviors investigated in the selected studies.

Type of Security Behavior	Study Focus	Frequency	Citation
Desirable security behavior	Compliance with policy/regulations	17	[25,49,57,61,63,64,65,68,69,71,73,75,77,78,83,84]
	Security protection	8	[25,49,62,63,69,70,72,78]
Undesirable security behavior	Risky security practices	4	[64,77,79,85]
	Violation/non-compliance	3	[50,68,82]
Both security behaviors	Secure and insecure practices	3	[66,76,83]

**Table 8 healthcare-10-02531-t008:** Individual factors as antecedents of security behavior.

Factor (n)	Key Points	User	DSB Study	USB Study
Self-efficacy (12)	Belief about self-capabilities to perform security practices	CS NS	[1,51,57,61,62,63,70,71,72,74,75,76]	N/A
Perceived severity (10)	Perception of adverse impacts from security incidents or threats	CS NS MS	[62,63,65,71,73,74,75,81]	[68,76]
Attitudes (7)	Positive or negative feelings about engaging in a specific behavior	CS NS	[51,58,60,62,70,80,81]	N/A
Subjective norms (7)	Perception of referent approval to exhibit or not exhibit a behavior	CS NS	[57,60,62,70,71,80,81]	N/A
Information security awareness (7)	Knowledge and understanding of health information security	CS NS MS	[57,65,76]	[64,76,77,83]
Perceived benefit/response efficacy (6)	Perception of positive outcomes from employing information security measures	CS NS	[62,71,74,75]	N/A
Perceived susceptibility/vulnerability (4)	Perception of the probability of being exposed to malicious threats	CS NS MS	[63,65,71,76]	N/A
Perceived behavioral control (4)	Perception of difficulty in displaying security behavior determined by internal or external constraints	CS NS	[60,62,70,80]	N/A
Perceived trust (4)	Belief that others’ actions can be instrumental to self-interest and provide benefits	CS NS	[1,51,60,74]	N/A
Perceived barriers (3)	Perception of the difficulty or cost of security practices, including money, time, or effort	CS NS	[72,74,75]	N/A
Perceived usefulness (2)	Protecting security and privacy is important and beneficial	CS NS	[70,81]	N/A
Perceived threat/risk (2)	Perceiving security threats as an inherent risk when using the HIS in a particular condition	CS NS	[63,65]	N/A
Safeguard cost (2)	Perception of inconvenience regarding the effort to employ security measures	CS NS	[63,65]	N/A
Perceived responsibility (2)	Personal characteristics prescribed in the code of ethics	CS MS	[62]	[79]
Personal norms (2)	Self-values and perspectives on information security	CS MS	[58]	[77]
Safeguard effectiveness (1)	Security safeguards can effectively mitigate the risks of utilizing the HIS in some circumstances	CS	[63]	N/A
Coping appraisal (1)	Examination of a person’s ability to deal with losses when faced with a threat	CS	[78]	N/A
Perceived work experience (1)	Perceptions of work experience that may help in enhancing information security competence and awareness	CS NS	[75]	N/A
Compatibility (1)	Perception of the protection is consistent with users’ needs, values, and experiences	CS	[70]	N/A
Controllability (1)	Perception of security measures can control the HIS	CS	[62]	N/A
Religion (1)	Religious values can influence perceptions and actions in protecting information security	CS NS	[57]	N/A
Personality traits (1)	Personality categories (e.g., extraversion, agreeableness, conscientiousness, neuroticism, intellect/imagination)	CS NS	[57]	N/A
Commitment (1)	Employee’s engagement to support information security in the organization	CS	[58]	N/A
Involvement (1)	Employee’s participation in supporting information security in the organization	CS	[58]	N/A
Perceived stress levels (1)	The mental state that can influence employees to use unfavorable security practices	CS NS	N/A	[85]
Situational empathy (1)	Personal characteristics in a situation that has sensitivity to the others’ emotional experiences to facilitate communication with patients and their families	MS	N/A	[79]
Perceived impact (1)	Impact levels of undesirable security practices that affect employees and others	MS	N/A	[79]
Self-control (1)	The process of self-regulation is such that the individual acts intentionally	MS	N/A	[77]
Lack of knowledge (1)	The employee does not have adequate knowledge of security requirements	NS	N/A	[50]
Poor skills (1)	The employee does not have adequate skills to carry out information security protection	NS	N/A	[50]
Poor discipline (1)	The employee does not have good discipline, e.g., laziness, arrogance, and indifference	NS	N/A	[50]

Notes: DSB = desirable security behavior (such as compliance behavior, protection behavior, etc.); USB = undesirable security behavior (such as risk-taking behavior, non-compliance, etc.); N/A = not applicable (no selected studies using the factor); CS = clinical staff; NS = non-clinical staff; MS = medical student.

**Table 9 healthcare-10-02531-t009:** Organizational factors as antecedents of security behavior.

Factor (n)	Key Points	Organization	DSB Study	USB Study
Organizational/management support (4)	Top-level management or organizational commitment to protecting information security	HS AHF NHF	[1,26,61,74]	N/A
Cues to action (3)	Information security campaigns and other influences that can encourage proper security behavior	HS AHF	[62,72,75]	N/A
Organizational culture/climate (3)	Multidimensional construct with numerous features that might influence employee behavior	HS NHF	[66,82]	[66]
Punishment/detection certainty (2)	Act or process certain to be enforced in data protection within the organization	HS NHF	[73,80]	N/A
Peer influence (2)	Influence from coworkers who have the power to give rewards or impose penalties for security practices	HS	[70]	[82]
Superior influence (2)	Influence from superiors who have the power to give rewards or impose penalties for security practices	HS	[70]	[82]
Facilitating condition (1)	Assets in ensuring that privacy protection behaviors are consistent with existing assets in the organization	HS	[70]	N/A
Data collection (1)	Techniques used for data collection become patient privacy concerns	HS	[69]	N/A
Secondary use (1)	Information is collected from the individual for a specific purpose but is used for another without proper authorization	HS	[69]	N/A
Error (1)	Intended and unintended errors in information collected by the organization	HS	[69]	N/A
Incentives (1)	Monetary and non-monetary incentives as a motivational stimulant	HS	[51]	N/A
Nature of work (1)	The quality of work done by staff	HS	[51]	N/A
Social relations (1)	Interpersonal connections among employees	HS	[51]	N/A
Security system satisfaction (1)	Degree of user satisfaction with the security system	HS	[76]	N/A
Legal/punishment (1)	Legal consequences or punishment from the organization for employees who conduct security violations/non-compliance	HS	[57]	N/A
Internal auditing effectiveness (1)	Procedures to ensure that information security control complies with organizational requirements and related standards	HS NHF	[73]	N/A
Security education and training program (SETA) (1)	Program to provide information security knowledge/skills and inform about information security policy for health staff	HS NHF	[73]	N/A
Workload (1)	Employees’ amount of work, busyness, and pressure at work that might disrupt their compliance behavior	HS	[60]	N/A
Regulatory concerns (1)	The risk of violating regulations regarding security and privacy related to HIS use	HS NHF	[65]	N/A
Medical assessment (1)	The patient’s medical status should be reported to related parties	AHF	N/A	[64]
Shadow working process (1)	Security practices enable efficient working practices but are against the policy or even national laws	HS NHF	N/A	[83]
Organizational limitations (1)	Organizational conditions that might cause human error, such as high turnover, low morale, understaffing, and/or high workload	NHF	N/A	[50]
Inefficient business processes (1)	Inefficient workflow that might cause human error, such as redundancy, suboptimality, and/or bottlenecks	NHF	N/A	[50]
Poor monitoring and enforcement (1)	Ineffective security policy implementation, such as few incentives to comply or penalties for violations	NHF	N/A	[50]
Physical environmental limitations (1)	Inadequate physical environment to support security control, such as small rooms	NHF	N/A	[50]
Technological limitations (1)	Inadequate technology to support security control, such as outdated computer applications, slow networks, etc.	NHF	N/A	[50]

Notes: DSB = desirable security behavior (such as compliance behavior, protection behavior, etc.); USB = undesirable security behavior (such as risk-taking behavior, non-compliance, etc.); N/A = not applicable (no selected studies using the factor); HS = hospital; AHF = academic healthcare facilities; NHF = non-specific healthcare facilities (e.g., clinics, health centers, etc.).

## Data Availability

Search results are available from the authors.

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
