# Peer review of "Information Security Behavior in Health Information Systems: A Review of Research Trends and Antecedent Factors"

_healthcare, 2022, doi:10.3390/healthcare10122531_

Round 1

Reviewer 1 Report

Dear authors, thank you very much for your submission to Healthcare and for giving me the opportunity to review your manuscript on the antecedents of cybersecurity behavior in heath care facilities. The topic refers to a relevant issue and is suitable for the journal’s aims and scope. Your research question is fundamental and worth investigating. The paper is well-structured, well-drafted, and well-written. The methodology is suitable. In the following, I report on several issues that should and/or could be addressed to increase the manuscript quality:

Title and Abstract

  1. I would use “cybersecurity” in the title and keywords. I think this would increase readings and citations.
  2. “Four criteria are utilized for inclusion and four 14 for disqualification.” I think this can be removed as it is too detailed for an abstract and also does not provide any insight as log as the criteria are not mentioned.

Introduction

  1. Page 3, l. 105: “We found 2 articles“. Please write „two“ (also in similar cases in the remainder of the paper).
  2. L. 116: I would not propose but ask a research question.
  3. The sections 3.1 to 3.3 address aspects that are not covered by your research question. I would therefore add further research questions. Maybe even your title should become broader.

Methodology

  1. In section 2.3, I think it would be useful if you add that you collected the mentioned data in a table.
  2. Also there, maybe add the notion of “cross referencing”.

Results

  1. Table 4: I would not consider literature reviews as “qualitative research”.
  2. I think the paper would benefit from a summarizing Figure in which all antecedents can be seen at one glance.

Discussion

  1. Please use a formulation, such as “The theoretical contributions of our research…” as a start of a paragraph dealing with these discussion-typical issues.
  2. Please use a formulation, such as “The practical implications of our research…” as a start of a paragraph dealing with these discussion-typical issues.
  3. Please use a formulation, such as “The limitations of our research…” as a start of a paragraph dealing with these discussion-typical issues.
  4. What future research do you recommend?

I hope you find my comments helpful. Good luck with your revision!

Author Response

Comments and Suggestions for Authors

Dear authors, thank you very much for your submission to Healthcare and for giving me the opportunity to review your manuscript on the antecedents of cybersecurity behavior in heath care facilities. The topic refers to a relevant issue and is suitable for the journal’s aims and scope. Your research question is fundamental and worth investigating. The paper is well-structured, well-drafted, and well-written. The methodology is suitable. In the following, I report on several issues that should and/or could be addressed to increase the manuscript quality:

Authors' Response: 

Thank you so much for your kind and insightful remarks on our recent article. They helped us enhance the paper. We did our best to take all the comments seriously. We trust that the quality of this improved version will match the standards of this reputable journal.

We have greatly enhanced the wording by using proofreading services. Following are the changes we made in response to the comments.

Title and Abstract

1. I would use “cybersecurity” in the title and keywords. I think this would increase readings and citations.

Authors' Reponse: We prefer the term “information security” rather than “” to coverage the wider scope of health information system implementation. We added explanation about the difference of “information security” and “cybersecurity” in l.115-121 as follows:

“In the literature of this research topic, the terms "information security" and "cybersecurity" are frequently used synonymously. Cybersecurity is related to the data in the cyberspace, in contrast to information security, which is the protection of all information [52]. In smaller healthcare facilities, it is possible that HIS implementation will not always be online. HIS security risks include medical staff members directly disclosing patient information to their families. Therefore, this study concentrated on information security behavior.”

 2. “Four criteria are utilized for inclusion and four 14 for disqualification.” I think this can be removed as it is too detailed for an abstract and also does not provide any insight as log as the criteria are not mentioned.

Authors' Response: We removed the sentence from abstract.

Introduction

3. Page 3, l. 105: “We found 2 articles“. Please write „two“ (also in similar cases in the remainder of the paper).

Authors' Response: We changed all one-digit number into text format.

4. L. 116: I would not propose but ask a research question.

Authors' Response: We changed the sentence into “Specifically, we asked the following research question...” (l.124)

5. The sections 3.1 to 3.3 address aspects that are not covered by your research question. I would therefore add further research questions. Maybe even your title should become broader.

Authors' Response: We changed the research questions as follows:

We also change the title into: “Information security behavior in health information system: A review of research trends and antecedent factors“

Methodology

6. In section 2.3, I think it would be useful if you add that you collected the mentioned data in a table.

Authors' Response: We add a sentence: “The results of the selected studies will be summarized in a table (File S2: Summary of selected studies)” in l.188-189.

7. Also there, maybe add the notion of “cross referencing”.

Authors' Response: We conducted our study using PRISMA framework and we did not use cross referencing.

Results

8. Table 4: I would not consider literature reviews as “qualitative research”.

Authors' Response: We excluded “literature review” study from “qualitative research” and make a new row in Table 4. (l.256)

9. I think the paper would benefit from a summarizing Figure in which all antecedents can be seen at one glance.

Authors' Response: We as summarizing of all antecedents’ factors to security behavior (l. 503)

Discussion

1. Please use a formulation, such as “The theoretical contributions of our research…” as a start of a paragraph dealing with these discussion-typical issues.

Authors' Response: We added the sentences “The theoretical contributions of our research…” in l. 505 to start the discussion about theoretical contributions.  

2. Please use a formulation, such as “The practical implications of our research…” as a start of a paragraph dealing with these discussion-typical issues.

Authors' Response: We added the sentences “The practical implications of our research…”in l.519 to start the discussion about practical implications. We also added some practical aspect to improve information security protection in the following sentences in that paragraph (l. 519-530)

3. Please use a formulation, such as “The limitations of our research…” as a start of a paragraph dealing with these discussion-typical issues.

Authors' Response: We add the sentences “The limitations of our research…“ in l.531

4. What future research do you recommend?

Authors' Response: We highlight future research direction in l. 541-544:

“Further research is necessary to learn more about the influencing factors among user groups in various health care organizations. Patients should be involved as research objects to determine how health care facilities should involve them in controlling information security.”

Reviewer 2 Report

Dear Author

Thank you for submitting your paper "Antecedent factors of information security behavior in health care facilities: a systematic review" to this esteemed journal "Healthcare (ISSN 2227-9032)". I read your paper and gave my concern down here:

1.  The title seems good. But, the contents the authors mentioned do not reflect the title.
2. The introduction section needs rework as I hardly find the motivation and the rationale of the study. The author must show the rationale of the study.
3. The paper seems to be more related to a Bibliometric review than a systematic review.
4. The authors mentioned in the title "Antecedent factors of information security behavior in health care facilities: a systematic review" but the contents do not reflect it. I thought that the authors will show a model showing the antecedents after the literature review. But, I found that they presented graphs, journal lists, authors...etc. 
5. This manuscript does not reflect the flow of SLR. 
6. The authors might mention the contributions of this paper or the originality of the study before the conclusion section.

Wish you all the best.

Author Response

Comments and Suggestions for Authors

Dear Author

Thank you for submitting your paper "Antecedent factors of information security behavior in health care facilities: a systematic review" to this esteemed journal "Healthcare (ISSN 2227-9032)". I read your paper and gave my concern down here:

Authors' Response:

Thank you so much for your kind and insightful remarks on our recent article. They helped us enhance the paper. We did our best to take all the comments seriously. We trust that the quality of this improved version will match the standards of this reputable journal.

We have greatly enhanced the wording by using proofreading services. Following are the changes we made in response to the comments.

1.  The title seems good. But, the contents the authors mentioned do not reflect the title.

Authors' Response: We changed the title into: “Information security behavior in health information systems: A review of research trends and antecedent factors

2. The introduction section needs rework as I hardly find the motivation and the rationale of the study. The author must show the rationale of the study.

Authors' Response: We explained the rationale of the study in Introduction section lines 100 – 112.

3. The paper seems to be more related to a Bibliometric review than a systematic review.

Authors' Response: We conducted our study according to PRISMA framework that are well known to do the systematic literature review. We explained it in l. 127-135 as follows:

To answer the research issue, we adopted a systematic literature review methodology. To conduct and report our review, we used the Preferred Reporting Items for Systematic Reviews and Meta-Analysis (PRISMA) statement [53]. PRISMA emphasizes on methods through which researchers may guarantee the transparent and thorough reporting of systematic reviews [54]. PRISMA 2020 updates the PRISMA 2009 statement, which included 27-item checklists, a flow diagram, and explanation [53]. The choice of a systematic review will provide us the opportunity to inquire into present trends in the emphasis placed on security behavior, security threats, and the variables that affect how users behave while protecting health information.

4. The authors mentioned in the title "Antecedent factors of information security behavior in health care facilities: a systematic review" but the contents do not reflect it. I thought that the authors will show a model showing the antecedents after the literature review. But, I found that they presented graphs, journal lists, authors...etc. 

Authors' Response: We added Figure 4 as summarizing of all antecedents factors to information security behavior in health care organization (l.503).

5. This manuscript does not reflect the flow of SLR. 

Authors' Response: We followed the PRISMA workflow (Figure 1) as a protocol of systematic literature review. We also attached PRISMA checklist in the Supplementary File S2.

6. The authors might mention the contributions of this paper or the originality of the study before the conclusion section.

Authors' Response:  We discussed theoretical contributions in the l. 505-518 and added practical contribution in l.519 – 530.

Round 2

Reviewer 2 Report

Congratulations